# Long Non-Coding RNAs as Strategic Molecules to Augment the Radiation Therapy in Esophageal Squamous Cell Carcinoma

**DOI:** 10.3390/ijms21186787

**Published:** 2020-09-16

**Authors:** Uttam Sharma, Tushar Singh Barwal, Varnali Acharya, Karuna Singh, Manjit Kaur Rana, Satyendra Kumar Singh, Hridayesh Prakash, Anupam Bishayee, Aklank Jain

**Affiliations:** 1Department of Zoology, Central University of Punjab, Bathinda 151 001, Punjab, India; uttamsharma1994@gmail.com (U.S.); tushar101singhbarwal@gmail.com (T.S.B.); acharya.varnali@gmail.com (V.A.); 2Department of Radiotherapy, Advanced Cancer Institute, Bathinda affiliated with Baba Farid University of Health Sciences, Faridkot 151 203, Punjab, India; karuna.mamc@gmail.com; 3Department of Pathology and Laboratory Medicine, All India Institute of Medical Sciences, Bathinda 151 001, Punjab, India; drmrsmanjitkaur@gmail.com; 4Centre for Advanced Research, King George’s Medical University, Lucknow 226 003, Uttar Pradesh, India; satsaiims@gmail.com; 5Amity Institute of Virology and Immunology, Amity University, Noida 201 313, Uttar Pradesh, India; hridayesh.prakash@gmail.com; 6Lake Erie College of Osteopathic Medicine, Bradenton, FL 34211, USA

**Keywords:** radioresistance, radiotherapy, radiosensitivity, esophageal squamous cell carcinoma, long non-coding RNAs

## Abstract

Intrinsic resistance to ionizing radiation is the major impediment in the treatment and clinical management of esophageal squamous cell carcinoma (ESCC), leading to tumor relapse and poor prognosis. Although several biological and molecular mechanisms are responsible for resistance to radiotherapy in ESCC, the molecule(s) involved in predicting radiotherapy response and prognosis are still lacking, thus requiring a detailed understanding. Recent studies have demonstrated an imperative correlation amongst several long non-coding RNAs and their involvement in complex cellular networks like DNA damage and repair, cell cycle, apoptosis, proliferation, and epithelial-mesenchymal transition. Additionally, accumulating evidence has suggested abnormal expression of lncRNAs in malignant tumor cells before and after radiotherapy effects in tumor cells’ sensitivity. Thus, lncRNAs indeed represent unique molecules that can influence tumor cell susceptibility for various clinical interventions. On this note, herein, we have summarized the current status of lncRNAs in augmenting resistance/sensitivity in ESCC against radiotherapy. In addition, we have also discussed various strategies to increase the radiosensitivity in ESCC cells under clinical settings.

## 1. Introduction

Globally, esophageal cancer (EC) is one of the most lethal cancer types and is the eighth most common cancer [1]. Esophageal squamous cell carcinoma (ESCC) accounts for almost 80% of all EC cases globally with an improper disposition and a higher incidence in the Asian ethnic population [2,3]. Plaintively, to the conventional notion, the current standard treatment strategies employed for metastatic ESCC patients are concurrent definitive chemo and radiotherapy. However, therapy resistance and tumor recurrence by radioresistant tumor cells are the major obstacles for ESCC treatment contributing to poor prognosis and dismal survival in ESSC patients (<20%) [4,5]. Furthermore, effective radiation-based treatment against radioresistant tumors involves higher radiation doses causing undesired side effects and damaging healthy tissue via “bystander effects” [6,7]. Therefore, identification of novel predictive and prognostic biomarkers is urgently needed to manage patients undergoing radiotherapy treatment.

Several studies have demonstrated the association of non-coding RNAs in regulating various cancer hallmarks, including the radioresistance processes in cells [8,9]. Our previous study suggested several microRNAs (miRNAs) dictating sensitivity and resistance against radiotherapy EC patients [10]. Furthermore, recent studies have reported the association of long non-coding RNAs (lncRNAs) with altered patients’ responses towards radiation therapy due to their active involvement in DNA damage response (DDR), apoptosis, and cell cycle arrest [11,12].

With this idea, we have collated scientific literature correlating several dysregulated lncRNAs corresponding to ESCC diagnostic when exposed to radiation therapy. Several long non-coding RNAs (Table 1) impede radiation therapy in ESCC by regulating the expression of several miRNAs modulating downstream targets involved in DDR, base excision repair pathway, ataxia telangiectasia mutated (ATM) pathway, and the respective DNA repair pathways. Additionally, another set of lncRNAs, such as *FAM201A,* sensitize the cancer cells towards radiation therapy in ESCC via targeting cyclin-dependent kinase subunit 1 (Cks1), cyclin D1, Mammalian target of rapamycin (mTOR) and activating Non-homologous end joining (NHEJ) pathway [13] thereby enhancing the radiosensitivity of cancer cells.

Herein, we have discussed several lncRNAs associated with the radioresistance and radiosensitivity of ESCC. We have also highlighted the plausible molecular pathways in decoding the radioresistance/sensitivity of ESCC. Furthermore, no review articles or meta-analysis were reported to date in lncRNAs and ESCC radiotherapy.

## 2. Literature Search and Selection Methodology

We used preferred reporting items for systematic reviews and meta-analysis (PRISMA) [23] for systematic analysis of the pieces of literature for this work. The literature search was performed by two individual authors (US and TSB) and further consulted with another researcher (AJ), whenever necessary. We considered the literature from January 2015–August 2020 to provide the updated information to the readers. We performed primary literature searches through various online databases, such as PubMed, ScienceDirect, and Google Scholar. We used a combination of several search keywords such as (“esophageal squamous cell carcinoma” OR “oesophageal squamous cell carcinoma” OR “ESCC”) AND (“radioresistance” OR Radio-resistance”) AND (“long non-coding RNA” OR “lncRNA” OR “Long noncoding RNA”); (“esophageal squamous cell carcinoma” OR “oesophageal squamous cell carcinoma OR “ESCC”) AND (“radio-resistance” OR “radio-resistance” OR “radio resistance”) AND (“lncRNA” OR “Long noncoding RNA” OR “Long non-coding RNA”) NOT (“miRNA” OR “micro RNA” OR “mi-RNA”). Furthermore, we were able to collate a total of (*n* = 236) pieces of literature from our primary literature search. A secondary search was subsequently performed post duplicate removal based on bibliographic data and a total of (*n* = 79) articles selected for final screening. After a careful scientific assessment, a total of (*n* = 11) articles were found suitable for further consideration eliminating book chapters, reviews, and conference abstracts (Figure 1).

As per the PRISMA analysis, various long non-coding RNAs involved in ESCC radioresistance (Section 3) and radiosensitivity (Section 4) are discussed below.

## 3. Detailed Regulation of Radioresistance in ESCC through lncRNAs

### 3.1. Dynamin 3 Opposite Strands

Dynamin 3 Opposite Strand *(DNM3OS*) spans ~7.9 kb on chromosome 1q.24.3 and transcribes in an antisense orientation of the 14th intron of the *DNM3* gene [24]. The *DNM3OS* was found to be expressed in a stage-dependent manner during several fetal development processes, such as the nasal process, pharyngeal arches, limb buds, and somites [24]. Furthermore, dysregulated *DNM3OS* is implicated in several cancers, including gastric cancer [25], ovarian cancer [25], and esophageal squamous cell carcinoma [12].

Zhang et al. [12] observed an upregulated expression (~6.30-fold) of *DNM3OS* in ESCC patients tissues (*n* = 16) compared to the matched healthy esophageal epithelial tissues. In addition, the upregulated expression levels of *DNM3OS* are associated with higher tumor grades. Additionally, the elevated expression of *DNM3OS* was also observed in several esophageal cancer cell lines viz. KYSE-30 (~39.40-fold), KYSE-150 (~35.80-fold), KYSE-180 (~28.50-fold), compared to the normal esophageal epithelial cell line Het-1A. To unveil the role of *DNM3OS* as a predictive biomarker for radiotherapy, the investigators irradiated ESCC cells (KYSE-150 and KYSE-150R) with 12 Gy radiations. They obtained a ~4.30-fold increase in the expression levels of *DNM3OS* in radioresistant KYSE-150R cells compared to parental cell KYSE-150 [12]. Furthermore, *DNM3OS* loss-of-function in radioresistant KYSE-150R cells led to an increase in sensitivity towards radiotherapy treatment. Thus, in vitro data suggest a positive correlation between *DNM3OS* expression and increased radioresistance in ESCC cells. Consistent with the above observations, slower tumor growth was observed in KYSE-30 xenografts Balb/c nude mice transfected with si*DNM3OS* cells compared to the control group mice irradiation with 12 Gy. Moreover, the tumor inhibition rate was higher for the *DNM3OS* knockout mice group, compared to the control group mice (76.59% vs. 47.03%) [12].

Further, an upregulated expression of *DNM3OS* in KYSE-30 (~39.20-fold) and KYSE-140 (~38.30-fold) observed in condition-media containing cancer-associated fibroblasts (CAFs) compared to the cells cultured in the standard medium [12]. It is well known that CAFs promote tumor growth in the tumor microenvironment by altering the extracellular matrix [12,26]. In addition, the dose enrichment ratio in KYSE-30 and KYSE-40 cultured in CAF containing condition-media was 3.3015 and 1.8368, respectively, compared to the control (standard medium), which confirmed the association of CAFs in modulating the expression of *DNM3OS* expression and thereby conferring radioresistance in ESCC [12]. Furthermore, to elucidate the underlying pathways responsible for *DNM3OS* deregulation, a molecule called “crenolanib” identified with kinase inhibitor signaling. Crenolanib is known to inhibit platelet-derived growth factor α/β (*PDGFR α/β*) signaling by targeting PDGF itself, thereby having the potential to inhibit *DNM3OS*. Moreover, a higher expression of PDGF β-polypeptide (*PDGFβ*), and platelet-derived growth factor receptor β-polypeptide (PDGFβR) observed in ESCC cell lines (KYSE-30 and KYSE-140) cultured in CAF-containing condition-media as compared to the controls [12]. Their association with *DNM3OS* expression regulation was further analyzed by transfection with PDGFRβ/PDGFβ siRNA in KYSE-30 and KYSE-140 cells before the treatment with CAF-containing condition-media. The above data suggest that PDGFRB/PDGF signaling pathway is involved in radioresistance in ESCC [12]. Furthermore, chromatin immunoprecipitation (ChIP) assay confirmed that the transcription factor Forkhead box protein O1 (FOXO1, a downstream molecule of PDGFRβ/PDGF pathway, regulates the transcription of *DNM3OS* by binding to *DNM3OS* promoter region [12].

Since the alteration in DNA repair enzymes regulates cell fate in response to radiotherapy treatment, the same authors investigated the involvement of *DNM3OS* in DNA damage response after radiation treatment to ESCC cells. Mechanistically, the levels of double-strand break proteins, such as H2A histone family member X (γH2AX) protein and cleaved poly ADP ribose polymerase (PARP), raised in irradiated si*DNM3OS* transfected ESCC cells compared to control cells [12]. Furthermore, the levels of DNA repair enzymes, such as pATM, Rad50, phosphorylated checkpoint kinase 2 (pChk2), Ku80, meiotic recombination 11 homolog 1 (MRE1), Nijmegen breakage syndrome 1 (NBS1), DNA protein kinase (DNA-PKcs), decreases in irradiated si*DNM3OS* transfected ESCC cells [12] (Figure 2 and Table 1).

The above findings suggest that the *DNM3OS* in association with CAFs induced radioresistance via targeting *PDGFβ* and *FOXO1* genes in ESCC, which indicate the importance of *DNM3OS* in ESCC as a target molecule.

### 3.2. Long Intergenic Non-Protein Coding RNA 473

Long intergenic non-protein coding RNA 473 (*LINC00473*) is located on the long arm of chromosome 6 at 6q27 [27]. Several studies have reported the association of dysregulated *LINC00473* in head and neck squamous cell carcinoma (HNSCC) [28] and ESCC [14], enhancing tumor progression inhibiting radiation response. Unfortunately, limited information is available on the association of *LINC00473* with radioresistance in ESCC.

In this regard, Chen et al. [14] observed an upregulated expression of *LINC00473* (~2.60-fold) in ESCC tumor tissues (*n* = 96) compared to the adjacent healthy tissues. Similar results were validated in the ESCC cell lines (TE-1, EC9706, Eca109, and KYSE-450), where TE-1 (~6.20-fold) and EC9706 (~6.80-fold) ESCC cell line demonstrated a higher expression level of *LINC00473* compared to normal esophageal epithelial cell line Het-1A (Table 1). The role of *LINC00473* was confirmed by irradiating the ESCC cell lines with an increasing dose of X-ray radiation (0, 2, 4, and 8 Gy) [14]. The increased expression of *LINC00473* positively correlated with clinicopathological characteristics, such as tumor node metastasis (TNM) stage and poor prognosis of ESCC [14]. Moreover, the shRNA-mediated knockdown of *LINC00473* paired with an accumulated dose of irradiation caused a lower survival fraction in the ESCC cell lines (TE-1 and EC9706) as assessed using the colony survival assay [14] (Table 1). Furthermore, another group suggested *LINC00473* upregulation in ESCC cells KYSE-30 (~1.20-fold) and TE-5 (~2.2-fold) after 8-doses of X-ray irradiation of 6 Gy each was given every 3 h in a total period of one day compared to the control cells [15]. Furthermore, increased cell proliferation and colony formation potential were observed in ESCC cell lines KYSE-30 and TE-5 when treated with augmented radiation (0, 2, 4, 6, and 8 Gy) doses; this validates the pivotal association of *LINC00473* in radioresistance [15]. Mechanistically, the authors observed reduced expression levels of poly (ADP-ribose) polymerase (PARP) protein in ESCC cells irradiated with 4 Gy radiations compared to the control group [14]. The PARP protein plays an important role in repairing DNA damage caused by chemo/radiotherapeutic drugs in cancer [29]. Subsequently, several investigators recognized miR-374a-5p and miR-497-5p as critical downstream targets of *LINC00473*, downregulating the expression profile against miR-374a-5p and miR-497-5p via competitive sponging [14,15]. Validating the role of *LINC00473* in association with miR-374a-5p and miR-497-5p in ESCC radioresistance, these researchers observed enhanced cell viability, colony survival, PARP, and CdC25A protein expression levels in miR-374a-5p and miR-497-5p knockdown in ESCC cell [14,15]. These results demonstrated the oncogenic character of *LINC00473*, augmenting the radioresistance in ESCC cells.

Furthermore, Cheng et al. [14] verified potential downstream targets against miR-374a-5p, where the *SPIN1* gene competed with *LINC00473* for miR-374a-5p binding (Figure 2 and Table 1). Additionally, SPIN1 overexpression promoted proliferation and colony formation in vitro, and induced tumor formation and reduced apoptosis in nude mice through PI3K/AKT signaling pathways in ovarian cancer [30,31]. Likewise, depletion of *SPIN1* oncogene suppressed the proliferation and survival fraction of ESCC cells and reduced the PARP protein levels, while its overexpression reversed the effects [14].

Overall, the above findings suggest that *LINC00473* weakened radiation therapy’s effect in ESCC patients via the cumulative activity of miR-374a-5p/miR-497-5p/SPIN1/PARP1; therefore, *LINC00473* could act as a novel prognostic biomarker and a potential therapeutic target for ESCC.

### 3.3. Non-Coding RNA Activated by DNA Damage

NORAD is a non-coding RNA activated by DNA damage (earlier known as *LINC00657*) associated intergenic lncRNA located on the long arm of chromosome 20 at 20q11.23. It is evolutionarily conserved among species and ubiquitously expressed in human tissues [32]. Further, its inactivation leads to chromosomal instability in diploid human cell lines. It is ascertained that this lncRNA acts as a multivalent binding platform for the Pumilio-Fem3-binding factor (PUF) family of protein, which plays an essential role in cell cycle control and neuronal activity [32].

Sun et al. [16] by quantitative real-time polymerase chain reaction (qRT-PCR) observed that *LINC00657* was upregulated in ESCC tissues by ~1.30-fold and in ESCC cell lines Eca109 (~1.50-fold), and KY-SE (~1.40-fold) after treatment with graded X-ray irradiations of 0, 4, and 8 Gy (Table 1) [16]. Further, it was observed that depletion of *LINC00657* in ESCC cells rendered them more sensitive to the radiotherapy treatments, as indicated by the reduced bromodeoxyuridine/5-bromo-2’-deoxyuridine (BrdU+) cell fraction and attenuated proliferation and migration potential of ESCC cells [16] (Figure 2 and Table 1). Additionally, when the ESCC cells were irradiated with a dose of 8 Gy, *LINC00657* sponged the miR-615-3p activity and reduced its level in the ESCC cells [16]. Notably, miR-615-5p are involved in lung cancer [33], hepatocellular carcinoma [34], and pancreatic ductal adenocarcinoma [35]. Additionally, by utilizing bioinformatics analysis, the authors identified *JunB* as the downstream target genes of miR-615-3p. Furthermore an inverse relationship amongst miR-615-3p and JunB protein expression profiles observed using Western blot assays, which suggests that *LINC00657* could increase the expression of JunB through sponging miR-615-3p and promotes the radioresistance of ESCC [16] (Figure 2, Table 1). Moreover, Kaplan–Meier analyses of a cohort of 183 ESCC patients showed shorter overall survival in the JunB-high expression group patients when compared with JunB-low expression group patients [16].

From the above results, we conclude that *LINC00657* increases the proliferation and migration potential of ESCC cells through targeting miR-615-3p and JunB [16], and enhances the radioresistance in ESCC cells.

### 3.4. POU6F2 Antisense RNA 2

POU6F2 antisense RNA 2 (*POU6F2–AS2*) encoded by the p14.1 locus of chromosome 7 and explicitly implicated to enhance ESCC progression [17]. Recently, Liu et al. [17] reported that *POU6F2-AS2* is involved in ionizing radiation-induced DNA damage response in ESCC. Moreover, an upregulated expression of *POU6F2-AS2* in irradiated ESCC cell lines KYSE-140~5-fold) KYSE-510 (~6.1-fold), KYSE-30 (~2-fold), and KYSE-70 (~1-fold), compared to the control cells [17] (Table 1). Further, depletion of *POU6F2-AS2* levels increased the ESCC cell sensitivity towards ionizing radiation. Subsequent experimental evidence, such as RNA pull-down, mass spectrometry, Western blot, and immunoprecipitation assays, suggested that Y-box binding protein (Ybx1) acted as the downstream target of *POU6F2-AS2* [17]. Ybx1 is a chromatin-binding protein that regulates RNA/DNA binding events depending upon its localization and implicated in DNA damage repair. Moreover, knockdown of *POU6F2–AS2* inhibited the recruitment of Ybx1 to the promoters of *cyclin B1* (*CCNB1*) and *p53* gene as well as to the DNA damage sites, thereby, confirming a close relationship between *POU6F2-AS2* and DNA repair pathways [17] (Figure 2, Table 1). In addition, ChIP-seq indicated a reduction in the signal corresponding to the binding of Ybx1 with chromatin in response to reduced *POU6F2-AS2* expression.

Therefore, these results demonstrated that *POU6F2-AS2* is essential for Ybx1′s chromatin localization, which subsequently aids in the growth of ESCC cells after DNA damage, hence collectively enhancing their radioresistance.

## 4. Detailed Regulation of Radiosensitivity in ESCC through lncRNA4

### 4.1. Family with Sequence Similarity 201 Member A

Family with sequence similarity 201 member (*FAM201A*) is a 2.9 kb long-non protein-coding gene, with five splice variants present on the p13.1 region of chromosome 9 [13]. Despite lacking the protein-coding capacity, it plays an essential role in the pathogenesis of various diseases. For instance, several of its single nucleotide polymorphisms (SNPs) associated with both obsessive-compulsive disorders of Tourette’s syndrome [36]. Additionally, its role was observed in osteonecrosis of the femoral head (ONFH) [37] in colorectal cancer [38].

Recently, Chen et al. [13] found significant upregulation of *FAM201A* in ESCC radioresistant patients (*n* = 15) compared to radiosensitive (*n* = 20) patients. Further analyses suggested that ESCC patients belonging to the FAM-high expression group (*n* = 13, ∆Ct = 8.437) exhibited an inadequate short term response to radiotherapy and had a lower survival time, contrary to the FAM-low expression group (*n* = 22, ∆Ct = 6.155), whereas, neither of the groups correlated with either T or N stage of tumor stage [13]. Moreover, the authors observed reduced survival and increased apoptotic rates after graded doses of radiation (0, 2, 4, 6, 8, and 10 Gy) in Eca109 radiosensitive cells compared to the Eca109R radioresistant cells [13] (Table 1). Furthermore, siRNA-mediated knockdown of *FAM201A* exhibited decreased proliferation in ESCC cell lines Eca109 and Eca109R. However, when Eca109R cells transfected with *FAM201A* mimics, the authors did not observe any increased proliferation because the expression level of *FAM201A* was already at a higher level in Eca109R cells [10]. Moreover the knockdown of *FAM201A* in a xenograft tumor mouse model significantly blocked tumor growth with decreased tumor volume and weight, indicating that *FAM201A* could induce radiosensitivity both in vitro and in vivo in the model system [13].

It was further observed that *FAM201A* confers radiosensitivity in ESCC by sponging its putative downstream target miR-101, and by regulating the expression of *mTOR* and *ATM* gene. A previous study by Yan et al. [39] showed that miR-101 functioned as a tumor suppressor in ESCC and regulated the radiosensitivity by inhibiting the mTOR and ATM expression. However, Chen et al. [13] by luciferase reporter assays, observed that *FAM201A* mutants suppressed the expression of miR-101 in Eca109 cells, suggesting a negative correlation between *FAM201A* and miR-101. Additionally, *FAM201A* overexpression using *FAM201A* mimics significantly reduced the miR-101 expression levels and enhanced the mTOR and ATM protein levels in Eca109/Eca109R cells [13] (Figure 3, Table 1). These results suggest that the increased ATM and mTOR levels due to *FAM201A* upregulation decrease the apoptotic rate, possibly by enhancing DNA repair by homologous recombination repair pathway (HRR) and non-homologous end joining (NHEJ) pathways.

Altogether, *FAM201A* regulates ESCC radiosensitivity via targeting miR-101/ATM/HRR axes and miR-101/mTOR/NHEJ axes, which suggests the potential of *FAM201A* as a diagnostic/prognostic biomarker and an excellent therapeutic target for ESCC.

### 4.2. Metastasis-Associated Lung Adenocarcinoma Transcript 1

Metastasis-associated lung adenocarcinoma transcript 1 (*MALAT1*) is alternatively known as nuclear enriched abundant transcript 2, (*NEAT2*) is a ubiquitously expressed, 8.7 kb lncRNA that regulates alternative splicing and transcription functions in cells [40]. *MALAT1* has been implicated in various malignancies and confers radiosensitivity, including gastric cancer [41], cervical cancer **[42]**, and esophageal squamous cell carcinoma [18].

Li et al. [18] by qRT-PCR observed a downregulated expression of *MALAT1* in ESCC cells (EC9706 and KYSE-150) treated with ionizing radiation of 5 Gy for 8 h. Additionally, the overexpression of *MALAT1* in irradiated EC9706 and KYSE-150 cells reduced the apoptotic rate leading to enhanced cell viability, suggesting the role of *MALAT1* in reducing the radiosensitivity of ESCC cells [18]. The authors further constructed the ESCC tumor xenograft model using a subcutaneous injection of KYSE-150 cells into the BALB/c nude mice and irradiated for five days with a total dose of 10 Gy γ-radiation. As a result, reduced tumor volume observed in the xenograft model compared to the control group [18]. Studies by Yao et al. [19] also revealed a similar downregulated expression of *MALAT1* in ESCC cell lines Eca-109, and TE-1 irradiated with a treatment dose of 3.2 Gy/min. In addition, these cells showed reduced spheroid formation capacity, stemness marker expression (SOX2 and Nanog), aldehyde dehydrogenase 1 (ALDH1) activity, and migration potential ESCC.

In order to better understand the mechanisms of the *MALAT1* in regulating radiosensitivity in ESCC cells, the authors identified cyclin-dependent kinase subunit 1 (Cks1) and Yes-associated protein (YAP) oncogenic transcriptional factor as the potential targets of *MALAT1* (Figure 3, Table 1). The authors further demonstrated that the depletion of *MALAT1* in ESCC cells inhibited the Cks1 levels at both mRNA and protein levels. In addition, it also decreased the translational activity of YAP and reduced the expression levels of connective tissue growth factor thus, collectively, enhancing apoptosis and reducing the stemness, cell proliferation, and migration potential of the ESCC cells when subjected to radiation therapy [18,19,43] (Figure 3, Table 1).

The above data confirmed that *MALAT1* modulates radiosensitivity and stemness of cancer stem cells by regulating Cks1 and YAP expression levels in ESCC.

### 4.3. Tumor Suppressor Candidate 7 or Limbic System Associated Membrane Protein

Tumor suppressor candidate 7 (*TUSC7*) or limbic system associated membrane protein (*LOC285194*), a 2 kb long lncRNA, consists of four exons and is located on 3q13.31 [44].

Tong et al. [20] by qRT-PCR, examined the expression of *LOC285194* in 142 ESCC tissues and observed that the expression of *LOC285194* was significantly reduced (~1.10-fold) (Table 1) compared to adjacent healthy tissues. The investigators further analyzed the expression levels of *LOC285194* by qRT-PCR in ESCC cell lines. They observed downregulated levels of *LOC285194* in KYSE-30 (~2.0-fold), KYSE-510 (~1.7-fold), KYSE-109 (~2.0-fold) cells lines compared to a standard esophageal epithelial cell line, Het-1A [20] (Table 1). Moreover, the *LOC285194* low expressing group showed a positive correlation with increased tumor size, higher histologic grade, i.e., G2 and G3/G4, advanced TNM stage, enlarged lymph node metastasis and distant metastasis as compared to the group with a high expression of *LOC285194* [20].

Additionally, to investigate the effect of radiation doses on the ESCC patients, 55 patients were irradiated with a total radiation dose of 40 Gy which was administered in 20 fractions each of 2 Gy, for four weeks with five fractions per week, during the first cycle of chemotherapy [20]. As expected, the low *LOC285194* expression group was more resistant towards radiotherapy as inferred from a 15% pathCR rate contrary to the 57% pathCR rate in the high *LOC285194* expression group [20]. The above results indicate that *LOC285194* could be used as a biomarker for screening and treating ESCC patients before esophagectomy. Furthermore, to study the prognostic significance of *LOC285194* expression, Kaplan–Meier analysis was performed, which revealed that low expression of *LOC285194* was associated with wretched disease-free survival (DFS) and overall survival (OS) [20]. Meanwhile, multivariate analysis revealed that low expression of *LOC285194* and distant metastases were independent factors that affected DFS and OS [20]. However, the molecular mechanisms that influence the radiotherapy sensitivity caused by *LOC285194* dysregulation are not understood well. Therefore, these results suggest that the downregulated expression of *LOC285194* signifies a higher risk of disease recurrence and treatment failure.

### 4.4. Actin Filament Associated Protein 1

Actin filament associated protein 1 (*AFAP1-AS1*), a 6810 bp long lncRNA, is transcribed from the actin filament associated protein 1 (*AFAP1*) gene in the antisense orientation and is located in the 14p16.1 region of chromosome 14. *AFAP1* gene codes the protein responsible for the cross-linking of actin filaments [45].

Zhou et al. [21], screened 18 lncRNAs, implicated in esophageal cancer in association with chemoradiotherapy resistance using qRT-PCR. *AFAP1-AS1*, along with two other lncRNAs, was deregulated more than 2-fold in the paired cell lines KYSE-30 and its resistant counterpart KYSE-30R (exhibited cross-resistance to 5-FU by ~3-folds and paclitaxel by ~5-fold). A similar result was observed in the parental TE-10 and cisplatin-resistant TE-10R cell lines. Furthermore, 204 patients underwent a total radiation dose of 60–70 Gy, i.e., 1.8–2.0 Gy/day for five days/week. A total of 162 out of the 204 patients were selected for further investigation, amongst which 81 cases belonged to a high *AFAP1-AS1* expression group, and the other 81 cases belonged to the low *AFAP1-AS1* expression group. In another cohort of 48 patients, the expression of *AFAP1-AS1* upregulation of ~0.81-fold was observed compared to adjacent healthy tissues [21]. This upregulation further correlated with lymph node metastasis, distant metastasis, advanced clinical stage, and resistance to definitive chemoradiotherapy (dCRT). However, no clinicopathological feature was observed to be associated with dCRT response. A similar upregulation of about ~2.2–15-fold was found in the cell lines KYSE-30, KYSE-70, KYSE-150, KYSE-450, KYSE-510, and KYSE-10 as compared to the healthy esophageal mucosa cell Het-1A [21] (Table 1).

Additionally, statistical analysis revealed that *AFAP1-AS1* could efficiently distinguish between ESCC samples and healthy esophageal mucosa (79.4% specificity and 73.3% sensitivity) and could also detect early ESCC (stage I + II, *n* = 79) with a specificity and sensitivity of 92.3% and 44.6% respectively [21]. However, results corresponding to the Kaplan–Meier survival analysis showed that higher levels of *AFAP1-AS1* were negatively associated with progression-free survival (PFS) and OS before dCRT [21]. Consistently, multivariate analysis revealed the upregulation of *AFAP1-AS1* as an influential independent prognosis factor of inferior PFS but the most significant unfavorable prognostic factor of OS [21].

These composite results demonstrate that *AFAP1-AS1* could be a predictive marker of clinical outcomes in ESCC patients treated with dCRT.

### 4.5. Taurine-Upregulated Gene 1

lncRNA taurine-upregulated gene 1 (*TUG1*) is a 6.7 kb-long transcript with six reading frames and mapped to the chromosome region 22q12.2. It was first identified as an upregulated transcript in the developing retinal cells [46].

Wang et al. [22] demonstrated that *TUG1* significantly upregulated in radiosensitive ESCC tissues (~1.5-fold) and cell lines, such as EC9706 and KYSE-30 (-3.0-fold and 2.3-fold, respectively) to their resistant counterparts (Table 1). The depletion of *TUG1* by siRNA mediated knockdown resulted in suppressed proliferation, migration, invasion, and increased apoptosis of EC9706 and KYSE-30 cell lines. Additionally, it was observed that exposure of *TUG1*-depleted cells with 2 Gy irradiation inhibited their proliferation, colony-forming ability, and induced apoptosis. These observations collectively concluded that *TUG1* is associated with ESCC radiosensitivity and radiotherapy [22]. Furthermore, the authors constructed an in vivo model by establishing KYSE-30 xenografts in Balb/c nude mice to confirm their previous findings. Interestingly, reduced tumor volume, tumor growth, and tumor weight observed in response to *TUG1* knockdown in the KYSE-30 xenografts compared to the controls after irradiation of 2 Gy [22].

Mechanistically, hsa-miR-144-3p and hsa-miR-145-5p were found as the potential target of *TUG1* [22]. The authors further indicated by Luciferase reporter assays that miR-144-3p was indeed a downstream target of *TUG1*, where these two transcripts were inversely correlated as established by Pearson’s correlation coefficient. miR-144-3p acts as a tumor suppressor in ESCC, and depletion of miR-144-3p restored the effects of *TUG1* suppression on radiotherapy, consequently validating its role as a regulator of radiation [22] (Figure 3, Table 1). Additionally, met proto-oncogene (*MET*) harbored a binding site for miR-144-3p at its 3’-UTR region. [43]. Further, the expression levels of MET protein and its mRNA levels were decreased in the cell lines transfected with miR-144-3p mimics. However, miR-144-3p inhibitors restored the MET expression levels that were suppressed in response to si-*TUG1*, suggesting the existence of lnc*TUG1*/miR-144-3p/MET axis [22].

*c-MET* is a proto-oncogene that codes for the protein c-MET receptor tyrosine kinase (c-MET RTK). It expresses in the epithelial tissues of many organs during embryogenesis adulthood [47]. In addition, MET induced upregulation of epidermal growth factor (EGFR) and increased Akt phosphorylation, which further reflects a resistance effect on cancer radiotherapy [22] (Figure 3, Table 1). Consistent with these findings, knockdown of MET decreased EGFR and phosphorylated-Akt (p-Akt) protein levels, possibly inhibiting cell proliferation [43]. However, reduced tumor levels of Ki67 and lowest levels of MET and p-Akt observed in the 2 Gy radiation group depleted with si-*TUG1* [22].

These composite results indicate that lnc*TUG1* regulates the radiosensitivity of ESCC via the *TUG1*/miR-144-3p/MET axis and can utilize it as a potential biomarker for radiotherapy.

## 5. Conclusion and Future Perspectives

We summarize the current scientific insights into the association of lncRNAs with a response to radiotherapy and a predictive biomarker for radiation treatment in ESCC patients. Further, we observed that lncRNAs might also act as potential therapeutic targets because of their differential expression patterns in ESCC cancerous tissue/cell lines before and after radiation therapy. The lncRNAs *DNM3OS*, *LINC00473*, *LINC00657*, *POU6F2–AS2* found to be involved in radioresistance, while *FAM201A*, *MALAT1*, *LOC285194*, *AFAP1-AS1*, *TUG1* lncRNAs linked to radiosensitivity in ESCC. As the upregulation of *DNM3OS* and *POU6F2–AS2* expression enhances radioresistance in ESCC cells. However, knockdown of lncRNA *DNM3OS* and *POU6F2–AS2* increased the extent of radiosensitivity in ESCC cells, as demonstrated by higher expression of yH2AX and reduced levels of DNA repair proteins followed by impaired double-strand break (DSB) repair [17,48]. The reviewed literature data indicate the potential efficacy of *DNM3OS* and *POU6F2–AS2* as targets to improve the outcome of radiotherapy. We have also highlighted the cumulative association of miRNAs and lncRNAs in many aspects of cellular response to radiation therapy. For example, oncogenic *LINC00473* favors radiation therapy resistance combined with mir-374a-5p, followed by alteration in DNA damage repair protein [14]. Furthermore, collated functions of *LINC00657* and miR-615-3p in ESCC cells hamper the TGF signaling pathways and impair the cellular response to radiation [16]. On the other hand, oncogenic *FAM201A* and *TUG1* favor radiosensitivity by inhibiting miR-101 and miR-144-3p, respectively. These findings suggest lncRNA and miRNA’s functional relevance, which together possess great potential in radiation therapy. Moreover, the preclinical studies and increased success rates of nucleic acid therapeutics provide an opportunity to target lncRNAs-miRNA for cancer treatment [16].

Radiotherapy is a well-established noninvasive treatment modality in which many patients do respond and get treated. However, several factors, such as familial mutation, epigenetic changes, and other comorbid pathological conditions, increase the resistance in a large cohort of patients, which is alarming and needs more in-depth analysis for improving interventions. Given this, our study addresses this and highlights lncRNAs as one molecular signature attributing to radiotherapy’s increased resistance. We have also proposed that modifying the lncRNA signature can influence the sensitivity of refractory patients. Furthermore, our study emphasized the role of lncRNAs in facilitating radiation therapy and suggested that LncRNA based therapeutics hold tremendous potential in improving cancer-directed therapies. lncRNA based intervention is an emerging area and expected to be a critical factor in deciding treatment modality for refractory or highly aggressive tumors ranging from gastric to lung and many more.

With the advancement of RNA-based therapeutics delivery, new combined radio-RNA therapies can emerge [49,50]. However, there would be challenges for the effective delivery of RNAi to cells; thus, there would be obstacles in the clinical trials-based study. Overall, multicentric studies in a large number of patients’ samples are warranted to utilize the lncRNAs based therapeutics in ESCC cells.

## Figures and Tables

**Figure 1 ijms-21-06787-f001:**
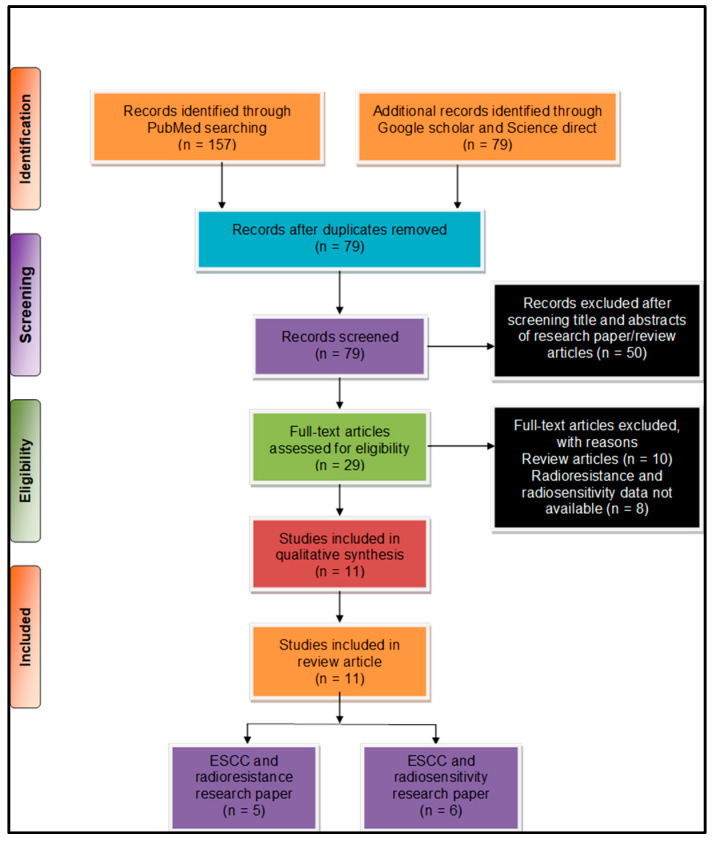
Preferred reporting items for systematic reviews and meta-analysis (PRISMA) flow chart describing the process of literature search and study selection related to esophageal squamous cell carcinoma (ESCC) and radioresistance/radiosensitivity. The total number of 11 relevant research articles were included in the review.

**Figure 2 ijms-21-06787-f002:**
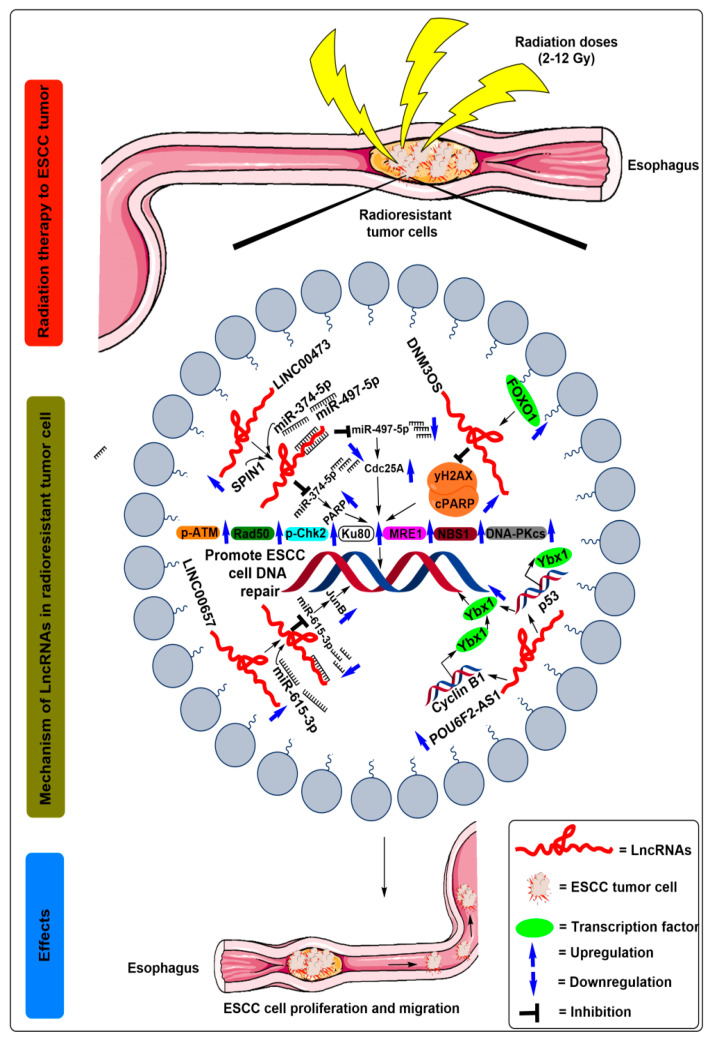
Schematic illustration of the molecular mechanisms of lncRNAs in the regulation of radioresistance in ESCC treatment. After radiation exposure to the ESCC cells, increased expression of FOXO1 transcribes the lncRNA *DNM3OS* by binding to its promoter region. Moreover, the increased expression of *DNM3OS* suppressed the levels of double-strand break proteins, such as H2A histone family member X (γH2AX) protein and cleaved poly ADP ribose polymerase (PARP) followed by increased levels of DNA repair enzymes, such as pATM, Rad50, phosphorylated checkpoint kinase 2 (pChk2), Ku80, meiotic recombination 11 homolog 1 (MRE1), Nijmegen breakage syndrome 1 (NBS1), DNA protein kinase (DNA-PKcs), and ultimately promote ESCC cell DNA repair. After radiation exposure to the ESCC cells, SPIN1, miR-374-5p, and miR-497-5p demonstrated competitive binding to upregulated lncRNA *LINC00473*. Moreover, miR-374-5p and miR-497-5p was suppressed after binding to the *LINC00473*, which is followed by the upregulated expression of PARP and Cdc25A, respectively, which are ultimately unable to break the double-strand DNA of ESCC cells. After radiation exposure to the ESCC cells, miR-615-5p competes for its binding to the upregulated lncRNA *LINC00657*. Moreover, the expression of miR-615-5p was suppressed after binding to the *LINC00657*, which is followed by the upregulated expression of JunB, which ultimately promotes the double-strand DNA repair of ESCC cells. After radiation exposure to the ESCC cells, POU6F2-AS1 expression was increased, which further recruits Ybx1 to the promoters of *cyclin B1* (*CCNB1*) and *p53* gene and the DNA damage sites and thus increases the Ybx1 protein levels, which ultimately promotes ESCC cell DNA repair.

**Figure 3 ijms-21-06787-f003:**
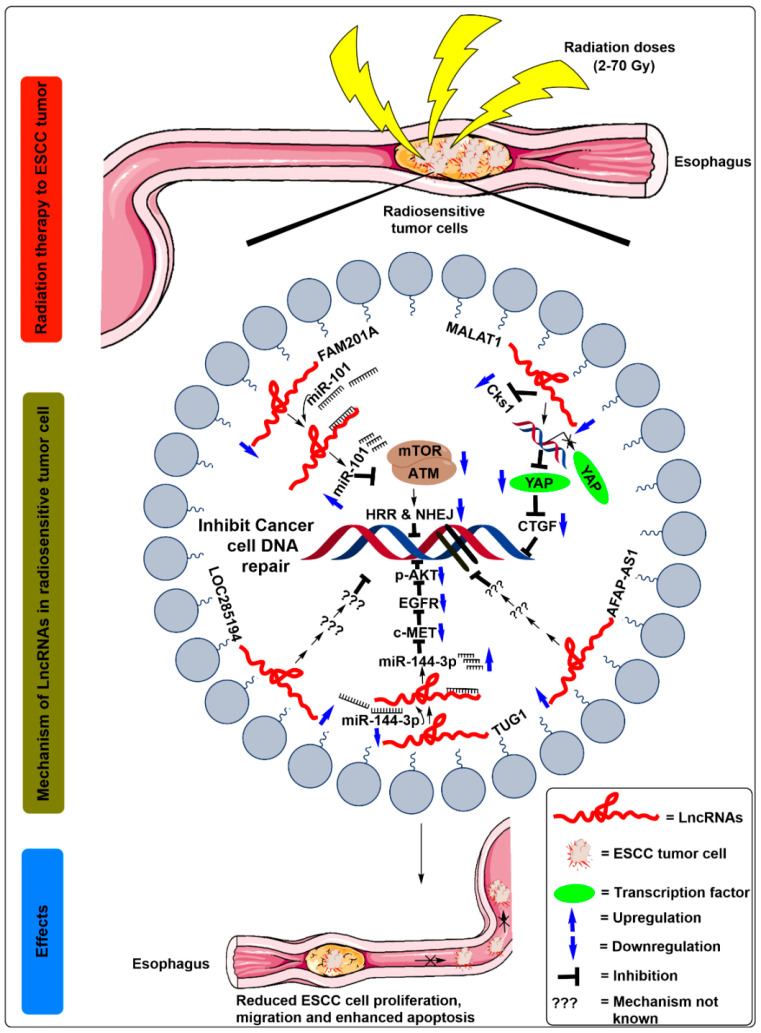
Schematic illustration of the molecular mechanisms of lncRNAs in the regulation of radiosensitivity in ESCC treatment. After radiation exposure to the ESCC cells, miR-101 binds to downregulate lncRNA *FAM201A*. Moreover, the expression of miR-101 was increased after binding to the *FAM201A*, which is followed by downregulated expression of mTOR and ATM, which decreases the homologous recombination repair (HRR) and non-homologous end joining (NHEJ) pathway and thus promotes the breakdown of double-strand DNA of ESCC cells. After radiation exposure to the ESCC cells, *MALAT1* expression was decreased, which further inhibits Cks1 levels at both mRNA and protein levels. In addition, it also decreased YAP’s translational activity, reducing the expression levels of connective tissue growth factor (CTGF), thus enhancing the breakdown of the double-strand DNA in ESCC cells. After radiation exposure to the ESCC cells, downregulated expression of lncRNAs *LOC285194* and AFAP-AS1 inhibits DNA repairing of ESCC cells. After radiation exposure to the ESCC cells, miR-144-3p competes with binding to the downregulated lncRNA TUG1. Moreover, the expression of miR-144-3p was increased after binding to the TUG1, which is followed by downregulated expression of c-MET, EGFR, and p-Akt protein, which ultimately promotes the breakdown of double strand DNA of ESCC cells and confers radiosensitivity.

**Table 1 ijms-21-06787-t001:** Characteristics of lncRNA in conferring radioresistance and radiosensitivity in ESCC.

lncRNA	Genomic Location	Mean Fold Change in Expression Compared to Controls	Radiation Type (Dose)	Property	Validation Methods	Biological Significance	Genes/Proteins/Pathways Affected	Ref.
*DNM3OS*	1q24.3	KYSE-30 ~39.4-folds KYSE-50 ~35.8-folds KYSE-180 ~28.5-folds KYSE-140 ~35-folds KYSE-450 ~7-folds Tissues ~6.3-folds KYSE-150R~4.3-folds	X-ray radiation (12 Gy, 12 days)	Oncogenic	qRT-PCR, Western blot analyses, ChIP assay	Increased levels of growth factors, reduced tumor suppressor levels, reduce DNA damage response leading to sustained proliferative signals, reduced apoptotic rate and enhanced radioresistance	*γH2AX* ↓*,* cPARP ↓, p-ATM ↑, Rad50 ↑, p-Chk2 ↑, Ku80 ↑, MRE1 ↑, NBS1 ↑, DNA-PK_cs_ ↑, and *p53* ↑, PDGFα/β signaling ↑	[12]
*LINC00473*	6q27	TE-1 ~6.2-folds EC9706 ~6.8-folds ECA109 ~3.6-folds KYSE-450 ~1.6-folds Tissues- ~2.6-folds	X-ray radiation (4 Gy)	Oncogenic	qRT-PCR, Western blot analyses	Reduced DNA damage response, reduced tumor suppressor levels, stimulated cell cycle progression leading to enhanced proliferation, reduced apoptotic rates, and enhanced radioresistance	mir-374a-5p ↓ PARP ↑ *SPIN1* ↑ miR-497-5p ↓ CdC25A ↑	[14,15]
*LINC00657*	20q11.23	KY-SE ~1.4-folds Eca-109 ~1.5-folds TE-1 ~1.4-folds Tissues ~1.3-folds	X-ray radiation (0, 4, 8 Gy)	Oncogenic	qRT-PCR, Western blot analyses, BrdU assay	Reduced tumor suppressor levels, causing increased cell proliferation, metastasis, invasion, and enhanced radioresistance	miR-615-3p ↓ JunB ↑ TGFβ ↑ TGF signaling ↑	[16]
*POU6F2-AS2*	7p14.1	KYSE-140 ~5-folds KYSE-510 ~6.1-folds KYSE-30 ~2.0-folds KYSE-70 ~1.0-fold	Ionizing radiation (0, 4, 6 Gy)	Oncogenic	qRT-PCR, microarray analyses, immunoblot, RIP assay, RNA pulldown assay, ChIP-seq assay	Reduced DNA damage response, reduced tumor suppressor levels, stimulated cell cycle progression leading to enhanced proliferation and enhanced radioresistance	Ybx1 ↑	[17]
*FAM201A*	9p13.1	NA	X-ray radiation (0, 2, 4, 6, 8 and 10 Gy)	Oncogenic	qRT-PCR, microarray analyses, Western blot analyses	Enhanced DNA damage response, reduced tumor suppressor levels leading to poorer radiosensitivity	miR-101 ↓ mTOR ↑ ATM ↑	[13]
*MALAT1*	11q13.1	Eca109 ~3.1-folds TE-1 ~3.4-folds	γ-radiation (5 Gy, 2.4 Gy/min) X-ray radiation (3.2 Gy/min)	Oncogenic	qRT-PCR, Western blot analyses, RIP assay	Enhanced invasion and metastasis, reduced radiosensitivity	Cks1 ↑, YAP ↑	[18,19]
*LOC285194*	3q13.31	KYSE-30 ~2.0-folds KYSE-510 ~1.7-folds KYSE-109 ~2-folds KYSE-70 ~1.1-folds KYSE-150 ~1.3-folds Tissues ~1.1-folds	X-ray radiation(40 Gy, 20 fractions of 2 Gy, 5 fractions/week for 4 weeks)	Oncogenic	qRT-PCR	Enhanced cell proliferation, enhanced apoptosis and reduced radiosensitivity	NA	[20]
*AFAP1-AS1*	14p16.1	KYSE-30 ~1.0-fold KYSE-70 ~15.0-folds KYSE-150 ~3.0-folds KYSE-450 ~5.0-folds KYSE-510 ~9.0-folds TE-10 ~3.0-folds Tissues ~0.8-folds	X-ray radiation(60–70 Gy, 1.8–2 Gy/day for 5 days/week)	Oncogenic	qRT-PCR	NA	NA	[21]
Lnc*TUG1*	22q12.2	EC9706 ~3.0-folds KYSE-30~2.9-folds KYSE-140 ~2.8-folds TE-13~2.8-folds	X-ray radiation (2 Gy)	Oncogenic	qRT-PCR, Western blot, RIP assay	Enhanced cell proliferation, invasion and reduced radiosensitivity	miR-144-3p ↓, MET ↑, p-Akt ↑	[22]

↑ = Upregulation; ↓ = Downregulation.

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
