# Peer review of "Long Non-Coding RNAs as Strategic Molecules to Augment the Radiation Therapy in Esophageal Squamous Cell Carcinoma"

_ijms, 2020, doi:10.3390/ijms21186787_

Round 1

Reviewer 1 Report

The review by Sharma et al. concerns the role of long non-coding RNAs in the response to radiation therapy of esophageal squamous adenocarcinoma. The topic is of interest, However the review is extremely difficult to read, and this strongly affects the understanding of the text (see for example point 4.4). The review consists of a description, in some cases very detailed, of the relevant works, but a personal view of the authors is lacking.
Even the two figures are difficult to understand, a legend with detailed explanations should be added.
I believe that the work should be revised as far as English is concerned and fundamentally rewritten.

Author Response

The authors of this manuscript express their sincere thanks to the reviewer for the critical assessment of this work. The authors have acted upon the recommendations of the reviewer which have resulted in a significant enhancement of the quality of this manuscript. The major modifications incorporated in the manuscript are highlighted using red color font. A “point-by-point” response to each and every comment is outlined below.

Suggestion 1 - The review by Sharma et al. concerns the role of long non-coding RNAs in the response to radiation therapy of esophageal squamous adenocarcinoma. The topic is of interest, However the review is extremely difficult to read, and this strongly affects the understanding of the text

Response - Thank you for your kind suggestions. We have made substantial changes to the manuscript as highlighted in red.

Suggestion 2 - The review consists of a description, in some cases very detailed, of the relevant works, but a personal view of the authors is lacking.

Response – We agree with your valuable suggestion. We have included our personal opinion on the study after each section of lncRNAs and in the conclusion and future perspective section highlighted in red.

Suggestion 3 - Even the two figures are difficult to understand, a legend with detailed explanations should be added.

Response – We agree with your valuable suggestion. We have included reworked figure legends in detail highlighted in red.

Suggestion 4 - I believe that the work should be revised as far as English is concerned and fundamentally rewritten.

Response – We agree with your valuable suggestion. We have done relevant revisions as per your suggestions.

Additionally,

  1. The reference list has been modified as we have added several new references. Special attention is given to conform to the order of references and bibliographic style of the journal.
  2. The entire manuscript has been thoroughly checked and edited to ensure uniform style, organization, and quality.

On behalf of my co-authors, I once again express my sincere thanks to the erudite reviewer for the valuable suggestions and constructive input to improve the quality of our manuscript.

Reviewer 2 Report

This review is interesting and may interest a large audience. However, it is difficult to read and requires extensive editing (I give few examples below but there are many more). In addition, an explanation and contextualization of the main concepts covered in the manuscript is necessary to make reading easier and therefore more attractive. Figure 2 and 3 as well as Table 1 are very important and very helpful.

-Authors: 3 authors with asterix but 2 are corresponding authors, it's confusing.

-It is not always clear what results are "in vitro" only (experiments) and which one are obtained from patient samples. So which one a really important in ESCC and which one are just working hypotheses (still important but some additional data are needed to ascertain their relevance in ESCC)?

  • Line 40-41: edit
  • Line 71: "we sincerely believe" this is not a motivation letter but a review;  the way the authors introduce their review is quite unusual but why not? still they don't need to write such sentence as if they wanted to convince the readers
  • Line 87-89: edit
  • Lack of transition (and give some context with some general scientific notions) between part 2 and part 3
  • Line 98-100: edit?
  • Line 100-101: edit
  • Line 103-105: edit
  • Line 122: edit
  • Line 152: "CAF-induced DNM3OS as a potential therapeutic target" and so what? how to target it? the authors need to give more details or better explain their statement
  • Line 157-158: edit?
  • Line 166-168: edit
  • Line 174-177: what do the authors mean?
  • Line 178-179: edit
  • Line 183-185: edit
  • Line 186-189: edit
  • Line 197-200: edit
  • Line 199: explain the "RUMILIO" family
  • Line 204-205: and so what? why is it important?
  • Line 208-210: edit
  • Line 212-214: this statement is too vague, it needs some explanation
  • Line 223-224: edit
  • Line 227:what "treated with a graded dose" mean?
  • Line 257-259: edit
  • Line 275-277: edit
  • Line 313-314: edit
  • Line 346-347: edit
  • Line 389-391: edit
  • Line 398-400: edit
  • Line 404-405: in ESCC!
  • Line 408-410: edit
  • Line 414-415: edit
  • Figure 2 and 3 legends need to be more detailed. There are a lot of informations, some are present within the main text of the manuscript but not all I think and it would be very useful for a reader to have a good grasp of the mechansims just by reading the legend and the figures.
  • One example, figure 3: a radiation dose of 70 Gy is HUGE, what is the context? is it cumulative?

Author Response

The authors of this manuscript express their sincere thanks to the reviewer for the critical assessment of this work. The authors have acted upon the recommendations of the reviewer which have resulted in a significant enhancement of the quality of this manuscript. The major modifications incorporated in the manuscript are highlighted using red color font. A “point-by-point” response to each and every comment is outlined below.

Suggestion 1 - This review is interesting and may interest a large audience. However, it is difficult to read and requires extensive editing (I give few examples below but there are many more). In addition, an explanation and contextualization of the main concepts covered in the manuscript is necessary to make reading easier and therefore more attractive. Figure 2 and 3 as well as Table 1 is very important and very helpful.

Response – We agree with your valuable suggestion. We have now taken care of this issue and highlighted corrections in red.

Suggestion 2 - Authors: 3 authors with asterix but 2 are corresponding authors, it's confusing.

Response - Thank you for your kind suggestion. We have now resolved the issue.

Suggestion 3 - It is not always clear what results are "in vitro" only (experiments) and which one are obtained from patient samples. So which one a really important in ESCC and which one are just working hypotheses (still important but some additional data are needed to ascertain their relevance in ESCC)?

Response – We agree with your valuable suggestion. We have discussed in the conclusion and future perspective section [Line numbers 434-436]

Suggestion 4 - Line 40-41: edit

Response - Thank you for your kind suggestion. We have now made substantial changes [Line numbers 41-42].

Suggestion 5 - Line 71: "we sincerely believe" this is not a motivation letter but a review; the way the authors introduce their review is quite unusual but why not? still they don't need to write such sentence as if they wanted to convince the readers

Response - Thank you for your kind suggestion. We have now made substantial changes [Line numbers 67-70].

Suggestion 6 - Line 87-89: edit

Response - Thank you for your kind suggestion. We have now made substantial changes [Line numbers 86-88].

Suggestion 7 - Lack of transition (and give some context with some general scientific notions) between part 2 and part 3

Response - Thank you for your kind suggestion. We have now added the connector between part 2 and part 3 or 4 [Line numbers 89-90].

Suggestion 8 - Line 98-100: edit?

Response - Thank you for your kind suggestion. We have now made substantial changes [Line numbers 98-99].

Suggestion 9 - Line 100-101: edit

Response - Thank you for your kind suggestion. We have now made substantial changes [Line numbers 99-101].

Suggestion 10 - Line 103-105: edit

Response - Thank you for your kind suggestion. We have now made substantial changes [Line numbers 102-103].

Suggestion 11 - Line 122: edit

Response - Thank you for your kind suggestion. We have now made substantial changes [Line numbers 119-122].

Suggestion 12 - Line 152: "CAF-induced DNM3OS as a potential therapeutic target" and so what? how to target it? the authors need to give more details or better explain their statement

Response - Thank you for your kind suggestion. We have now made substantial changes [Line numbers 148-150].

Suggestion 13 - Line 157-158: edit?

Response - Thank you for your kind suggestion. We have now made substantial changes [Line numbers 171-175].

Suggestion 14 - Line 166-168: edit

Response - Thank you for your kind suggestion. We have now made substantial changes [Line numbers 180-183].

Suggestion 15 - Line 174-177: what do the authors mean?

Response - Thank you for your kind suggestion. We have now made substantial changes [Line numbers 186-191].

Suggestion 16 - Line 178-179: edit

Response - Thank you for your kind suggestion. We have now made substantial changes [Line numbers 191-194].

Suggestion 17 - Line 183-185: edit

Response - Thank you for your kind suggestion. We have now made substantial changes [Line numbers 194-208].

Suggestion 18 - Line 186-189: edit

Response - Thank you for your kind suggestion. We have now made substantial changes [Line numbers 202-208].

Suggestion 19 - Line 197-200: edit

Response - Thank you for your kind suggestion. We have now made substantial changes [Line numbers 215-217]

Suggestion 20 - Line 199: explain the "RUMILIO" family

Response - Thank you for your kind suggestion. We have now made substantial changes [Line numbers 218-220].

Suggestion 21 - Line 204-205: and so what? why is it important?

Response - Thank you for your kind suggestion. We have now made substantial changes [Line numbers 234-236].

Suggestion 22 - Line 208-210: edit

Response - Thank you for your kind suggestion. We have now made substantial changes [Line numbers 226-228].

Suggestion 23 - Line 212-214: this statement is too vague, it needs some explanation

Response - Thank you for your kind suggestion. We have now made substantial changes [Line numbers 228-234].

Suggestion 24 - Line 223-224: edit

Response - Thank you for your kind suggestion. We have now made substantial changes [Line numbers 242-243].

Suggestion 25 - Line 227:what "treated with a graded dose" mean?

Response - Thank you for your kind suggestion. We have now made substantial changes [Line numbers 244-246].

Suggestion 26 - Line 257-259: edit

Response - Thank you for your kind suggestion. We have now made substantial changes [Line numbers 275-278].

Suggestion 27 - Line 275-277: edit

Response - Thank you for your kind suggestion. We have now made substantial changes [Line numbers 293-295].

Suggestion 28 - Line 313-314: edit

Response - Thank you for your kind suggestion. We have now omitted this line to make the sentence clear (Line numbers 342-343).

Suggestion 29 - Line 346-347: edit

Response - Thank you for your kind suggestion. We have now made substantial changes [Line numbers 374-377].

Suggestion 30 - Line 389-391: edit

Response - Thank you for your kind suggestion. We have now made substantial changes [Line numbers 418-420].

Suggestion 31 - Line 398-400: edit

Response - Thank you for your kind suggestion. We have now made substantial changes [Line number- 426-428].

Suggestion 32 - Line 404-405: in ESCC!

Response - Thank you for your kind suggestion. We have now made substantial changes. [Line numbers 433-444].

Suggestion 33 - Line 408-410: edit

Response - Thank you for your kind suggestion. We have now made substantial changes [Line numbers 434-439].

Suggestion 34 - Line 414-415: edit

Response - Thank you for your kind suggestion. We have now made substantial changes [Line numbes 442-444].

Suggestion 35 - Figure 2 and 3 legends need to be more detailed. There are a lot of information’s, some are present within the main text of the manuscript but not all I think and it would be very useful for a reader to have a good grasp of the mechansims just by reading the legend and the figures.

Response – We agree with your valuable suggestion. We have now mentioned the figure legends in detail (Line numbers 152-169 and 297-311).

Suggestion 36 - One example, figure 3: a radiation dose of 70 Gy is HUGE, what is the context? is it cumulative?

Response – We agree with your valuable suggestion. 70Gy is the cumulative dose mentioned in the figure.

Additionally,

  1. The reference list has been modified as we have added several new references. Special attention is given to conform to the order of references and bibliographic style of the journal.
  2. The entire manuscript has been thoroughly checked and edited to ensure uniform style, organization, and quality.

On behalf of my co-authors, I once again express my sincere thanks to the erudite reviewer for the valuable suggestions and constructive input to improve the quality of our manuscript.

Round 2

Reviewer 1 Report

The review has been substantially ameliorated, therefore I think it is now suitable for publication.

Reviewer 2 Report

Line 44: I am not familiar with the word "Plaintively"

Line 246: there is a blue box in the middle of the sentence